# Emergent Global OOD Performance in Multimodal Mammography Models

## Abstract

Out-of-distribution (OOD) generalization is a central barrier to deploying mammography AI fairly across health systems, especially in LMICs. We test whether parameter scale alone, within one architecture family and without adding extra images, yields emergent robustness in a CLIP-trained vision–language model (VLM). Using EMBED-open (22k patients; 480k 2D views), we train on cohort-1 only (11k patients; 240k images) a CLIP VLM whose ViT patch-16 image tower spans 6M, 22M, 86M, and 307M parameters; multimodality is limited to short, de-identified captions (sex, age, ethnicity, manufacturer, view) with a frozen biomedical text encoder, and downstream clinical targets are modeled via linear probes on learned embeddings. We evaluate zero-shot on five international OOD cohorts—DMID (India), VinDr-Mammo (Vietnam), KAU-BCMD (Saudi Arabia), CMMD (China), and INBreast (Portugal). In-domain performance increases roughly linearly with log-parameters, whereas OOD performance shows a modest upward trend with a pronounced knee at the largest scale on several cohorts; gains are most visible at the most clinically relevant low-FPR operating points (pAUC@0–10% and TPR@5% FPR). Achieved without extra data, task labels during training, or architectural changes, this pattern is consistent with emergent global OOD robustness from scale in multimodal mammography models. Consequently, parameter scale should be a first-class design lever for globally deployable VLMs, and downsizing via smaller models, pruning, or quantization should be accompanied by OOD monitoring across international cohorts to avoid eroding this scale-driven robustness.

## 1 Background

### 1.1 Mammography AI and the OOD challenge

Deep learning has achieved strong screening performance in mammography, but models often face distribution shifts across institutions, vendors, demographics and acquisition protocols. Early large-scale evaluations (, a UK/US multi-centre system) demonstrated competitive or superior accuracy to radiologists yet also highlighted site and prevalence differences that complicate deployment (McKinney et al., 2020). Risk-prediction work further underscored the need for robustness and calibration under cross-site validation (Yala et al., 2021). These observations motivate methods and datasets that enable rigorous, multi-domain assessment of both accuracy and safety under out-of-distribution (OOD) shifts.

### 1.2 EMBED: a large, diverse public resource for robustness studies

The EMory BrEast imaging Dataset (EMBED) was released to address scale and diversity gaps in public mammography corpora, providing 3.4M images from ∼116k patients with lesion annotations, pathology outcomes and rich acquisition metadata across 2-D DM, synthetic 2-D (C-view) and DBT (Jeong et al., 2023). Its openly accessible subset (*EMBED-open*) has catalyzed studies on density estimation, risk prediction and representation learning while enabling fairness and OOD analysis at scale.

### 1.3 CLINICALLY ORIENTED MODELING ON EMBED

On clinically proximate tasks, Donnelly et al. (2024) introduced an interpretable 1–5 year risk model that replaces complex reasoning with a bilateral-asymmetry module, retaining much of Mirai's performance while offering clearer explanations on EMBED and an external Duke cohort. For breast density, Khara et al. (2024) trained ResNet-34 models that generalized across FFDM and synthetic 2-D modalities and reported no substantial performance gaps between Black and White subgroups on EMBED-open, illustrating how dataset diversity supports equity analyses. Longitudinal risk modeling that explicitly aligns attention across prior and current exams (Wang et al. (2024)) further improved time-to-event prediction on EMBED, suggesting temporal structure is an important clinical prior.

### 1.4 VISION–LANGUAGE PRETRAINING AND ROBUSTNESS/OOD

Recent work adapts contrastive vision–language learning to mammography. Du et al. (2024a) showed that a parameter-efficient CLIP variant trained on EMBED captions and biomedical text can support zero-shot transfer (e.g., BI-RADS, density) while reducing trainable parameters. Du et al. (2024b) extended this line with multi-view, multi-scale alignment, reporting improved zero-shot and localization performance across EMBED and external cohorts. In parallel, robustness and OOD safety have been studied through fairness auditing and domain shift. Huang et al. (2024) re-trained CNN/Transformer backbones on EMBED-open and analyzed race/ethnicity performance gaps, finding that label re-weighting reduced disparities at a small cost to global AUROC. For cross-dataset adaptation on detection, Ashraf et al. (2024) proposed a mask-annealed student–teacher transformer that improved sensitivity on INBreast/CBIS-DDSM without target labels. Complementary directions include counterfactual contrastive augmentation for scanner/domain shift (Roschewitz et al., 2024), multi-modal system design spanning FFDM, synthetic 2-D and DBT (Park et al., 2025), and subgroup analyses of commercial DBT models across demographics and imaging factors (Harakeh et al., 2025).

### 1.5 GAP AND POSITIONING

Despite these advances, two gaps remain. First, most mammography OOD studies either (i) emphasize supervised clinical endpoints with task-specific architectures (Donnelly et al., 2024; Khara et al., 2024; Wang et al., 2024) or (ii) explore CLIP-style pretraining with additional modeling changes or captions beyond minimal acquisition/demographic text (Du et al., 2024a;b). Second, there has been limited isolation of *parameter scale* as a single experimental factor within one architecture family trained end-to-end on a fixed, single-institution source and then evaluated zero-shot across multiple international cohorts. Our study addresses this gap by holding data, objective, and multimodal text constant while sweeping model size in a CLIP-trained VLM, measuring in-domain trends and OOD behavior at clinically relevant low-FPR operating points across diverse external datasets.

## 2 METHODS

### 2.1 STUDY DESIGN AND DATASETS

We test whether *parameter scale alone* induces emergent OOD robustness when all other factors are held fixed. We train a CLIP-style vision–language model (VLM) on EMBED-open cohort-1 only (approx. 11k patients; ~240k 2D images) and use EMBED-open cohort-2 for in-domain (ID) linear-probe fitting and model selection. Zero-shot OOD evaluation is performed (no fitting on target data) on five international cohorts: DMID (India), VinDr-Mammo (Vietnam), KAU-BCMD (Saudi Arabia), CMMD (China), and INBreast (Portugal). Across all datasets we evaluate at the *image level*; when only exam-level labels exist, labels are broadcast to all images from that exam.

### 2.2 MINIMAL MULTIMODAL SUPERVISION

To constrain the language signal, we generate short, de-identified captions with a strict schema: `sex:<..>`, `age:<..>`, `ethnicity:<..>`, `manufacturer:<..>`, `image_view:<..>`. Demographic fields are parsed from EMBED tables; missing values are

set to `unknown`. We apply caption dropout (remove the `manufacturer:` token with probability 0.25 during training) to mitigate trivial site/vendor shortcuts. The text tower is a frozen biomedical encoder (Bio-ClinicalBERT), projecting the [CLS] embedding to a $d$=512 joint space; the text model is never fine-tuned.

## 2.3 Architecture family and scaling

The image tower is a ViT patch-16 backbone drawn from a single family with four sizes: XXS ($\sim$6M), S ($\sim$22M), M ($\sim$86M), L ($\sim$307M) parameters. Each feeds a linear projection to a $d$=512 unit-norm embedding. The CLIP logit scale is learned. No architectural or data changes are introduced across scales.

## 2.4 Training protocol (contrastive pretraining)

We use InfoNCE with symmetric image$\leftrightarrow$text losses. Images are resized to $384\times384$ and normalized with CLIP statistics. Augmentations emphasize domain/physics plausibility: mild geometry (random resized crop, small affine/perspective jitter; no flips to preserve laterality) followed by scanner-style photometric changes (gamma, piecewise histogram warp, CLAHE or equalize, blur/unsharp, downsample–upsample, JPEG, Gaussian/Poisson/speckle noise, vignette). Optimization uses AdamW with cosine decay, warmup in the first 2 epochs, gradient clipping, bfloat16 autocast, and (where supported) fused ops. We enable gradient checkpointing in the image tower. A FIFO contrastive queue (MoCo-style, $K$=4096) provides additional negatives after warmup. We target $\sim$5k optimizer updates, with early stopping based on cohort-2 validation. To reduce selection noise, we perform *endpoint ensembling* over epochs 16–20: an exponential moving average (EMA) of weights over the last 5 epochs for CMMD/DMID/KAU/INBreast and stochastic weight averaging (SWA) for VinDr. We report results using these fixed recipes: `ckpt_EMA_e16to20` (most cohorts) and `ckpt_SWA_e16to20` (VinDr).

## 2.5 Linear probe fitting and evaluation

After contrastive training, we freeze the image tower and extract per-image embeddings. We then fit a regularized logistic regression (L2) on EMBED-open cohort-2 (ID) to predict an actionable cancer label derived from clinical/pathology text or BI-RADS $\geq$4 (consistent across cohorts via a unified heuristic mapping). The same linear probe is applied *unchanged* to every OOD dataset (no calibration transfer or threshold tuning on target data).

## 2.6 Metrics

We report AUROC, **pAUC@0–10% (normalized)** and **TPR@5% FPR**. Normalized pAUC is computed by integrating the ROC curve over FPR $\in [0, 0.10]$ and dividing by 0.10, yielding a $[0, 1]$ score emphasizing clinically relevant low-FPR operation. TPR@5% FPR is the sensitivity obtained at the point on the ROC with 5% false-positive rate (computed on each dataset's ROC curve without target-domain fitting). All results are point estimates; we fix seeds across scales and do not perform hyper-parameter tuning per model size.

## 2.7 Compute

All experiments run on a single 80 GB NVIDIA A100 with $\geq$160 GB host RAM. Data loading uses persistent workers and prefetch; manifests and resolved file paths are cached to avoid repeated indexing in re-runs.

## 3 Results

### 3.1 In-domain scaling trends

On EMBED-open (ID), performance improves monotonically with model size: AUROC rises from **0.614** (XXS) $\rightarrow$ **0.659** (S) $\rightarrow$ **0.682** (M) $\rightarrow$ **0.697** (L). Low-FPR metrics show similar gains: pAUC@0–10% increases **0.129** $\rightarrow$ **0.153** $\rightarrow$ **0.187** $\rightarrow$ **0.197**, and TPR@5% FPR increases **0.133**

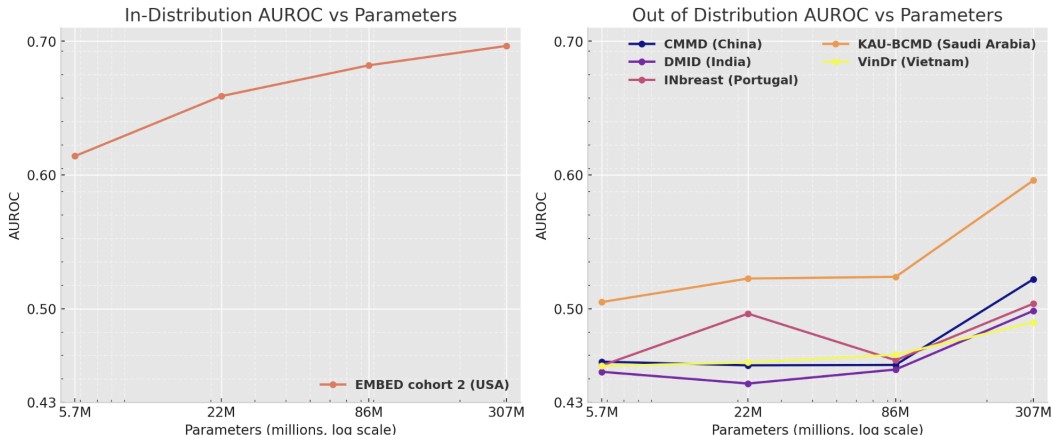

Figure 1: Scaling of classification AUROC with model size. In distribution (EMBED-open cohort-2) vs Zero Shot Out of Distribution performance. OOD performance elbow is visible at the largest model scale accross all OOD datasets tested.

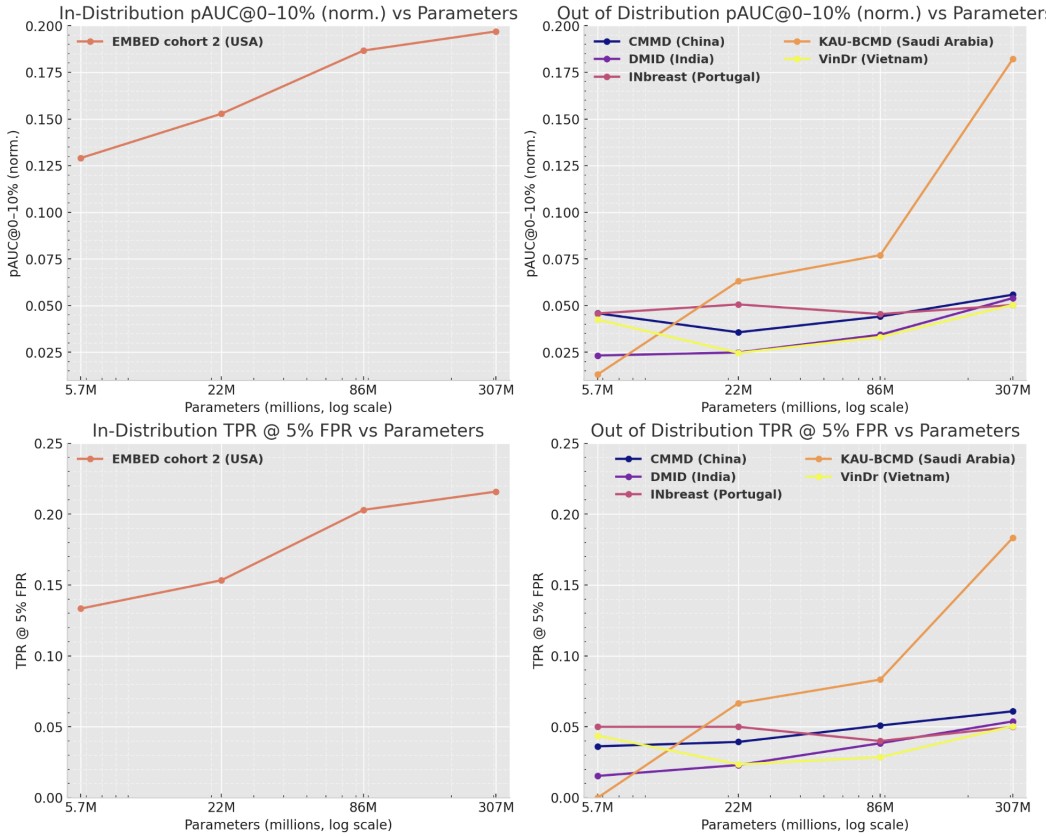

Figure 2: Low-FPR operating performance vs model size. 'Elbow' at the largest size is prominent for KAU-BCMD.

$\rightarrow$ **0.153** $\rightarrow$ **0.203** $\rightarrow$ **0.216**. These trends are consistent with a roughly linear improvement versus log-parameters under a fixed data/architecture regime.

| Split | Dataset | ViT-XXS | | | ViT-S | | | ViT-M | | | ViT-L | | |
|---|---|---|---|---|---|---|---|---|---|---|---|---|---|
| | | AUROC | pAUC | TPR@5% | AUROC | pAUC | TPR@5% | AUROC | pAUC | TPR@5% | AUROC | pAUC | TPR@5% |
| Out-of-Distribution | CMMD (CN) | 0.460 | 0.046 | 0.036 | 0.458 | 0.036 | 0.039 | 0.458 | 0.044 | 0.051 | **0.522** | **0.056** | **0.061** |
| | DMID (IN) | 0.453 | 0.023 | 0.015 | 0.444 | 0.025 | 0.023 | 0.455 | 0.034 | 0.038 | **0.498** | **0.054** | **0.054** |
| | INbreast (PT) | 0.458 | 0.046 | 0.050 | 0.496 | **0.051** | **0.050** | 0.461 | 0.045 | 0.040 | **0.504** | 0.050 | 0.050 |
| | KAU-BCMD (SA) | 0.505 | 0.013 | 0.000 | 0.523 | 0.063 | 0.067 | 0.524 | 0.077 | 0.083 | **0.596** | **0.182** | **0.183** |
| | VinDr (VN) | 0.457 | 0.043 | 0.044 | 0.460 | 0.025 | 0.024 | 0.466 | 0.033 | 0.029 | **0.490** | **0.050** | **0.051** |
| In-Distribution | EMBED-open (US) | 0.614 | 0.129 | 0.133 | 0.659 | 0.153 | 0.153 | 0.682 | 0.187 | 0.203 | **0.697** | **0.197** | **0.216** |

## 3.2 Zero-shot OOD performance across five cohorts

Across CMMD, DMID, INBreast, KAU-BCMD, and VinDr, AUROC values remain close to chance for small models but improve with scale. Averaged over the five OOD sets, AUROC increases from **0.467** (XXS) → **0.476** (S) → **0.473** (M) → **0.522** (L), a mean gain of $+0.055$ (approximately $+12\%$ relative). Low-FPR metrics show the clearest scale benefits: mean normalized pAUC@0–10% rises from **0.034** (XXS) → **0.040** (S) → **0.047** (M) → **0.079** (L), and mean TPR@5% FPR rises from **0.029** (XXS) → **0.041** (S) → **0.048** (M) → **0.080** (L). Relative to XXS, the L model improves mean pAUC by $\sim2.3\times$ and TPR@5% by $\sim2.7\times$.

**Cohort-specific patterns.** The largest gains appear on **KAU-BCMD**, where AUROC climbs from **0.505** (XXS) to **0.596** (L), and normalized pAUC exhibits a pronounced knee from **0.013** (XXS) to **0.182** (L), with TPR@5% FPR increasing from **0.000** to **0.183**. Other cohorts show smaller but consistent improvements at the largest scale: *CMMD* AUROC **0.460** → **0.522**, pAUC **0.046** → **0.056**; *DMID* AUROC **0.453** → **0.498**, pAUC **0.023** → **0.054**; *INBreast* AUROC **0.458** → **0.504** with pAUC near-tie between S (**0.051**) and L (**0.050**); *VinDr* AUROC **0.457** → **0.490**, pAUC **0.043** → **0.050**. These results were computed with a single linear probe trained on EMBED (ID) and applied unchanged to each OOD cohort.

**Operating-point sensitivity.** Scale effects are most visible at clinically relevant low-FPR regions: even where full AUROC shifts are modest, pAUC@0–10% and TPR@5% FPR improve substantially at L, suggesting *emergent* gains in the part of the ROC curve that matters for screening triage.

## 3.3 Checkpoint averaging

Using EMA over epochs 16–20 (`ckpt_EMA_e16to20`) stabilized ID selection and OOD transfer on four cohorts. VinDr benefited from SWA over the same window (`ckpt_SWA_e16to20`), yielding small but consistent pAUC/TPR gains versus single-epoch snapshots.

# 4 Discussion and Conclusion

## 4.1 Evidence for scale-driven, low-FPR OOD gains

Under a fixed architecture, objective, data domain and minimal captions, increasing ViT capacity yields: (i) monotonic ID improvements and (ii) *emergent* OOD gains that are most pronounced at low FPR. The knee-like jump on KAU-BCMD and the consistent mean improvements across all five cohorts (especially in pAUC and TPR@5% FPR) support the hypothesis that *scale alone* can unlock more transferable features in a CLIP-trained VLM even without task labels or extra pretraining data.

## 4.2 Why low-FPR metrics move more than AUROC

Screening deployment typically operates at stringent specificity levels. Contrastive pretraining aligns image and acquisition/demographic text; larger image towers appear to reduce representation noise near the decision boundary, improving sensitivity at fixed low FPR before noticeable shifts in global AUROC. This is desirable in clinical settings and suggests that OOD evaluations should report pAUC@0–10% and TPR@5% FPR alongside AUROC.

### 4.3 Controls we did *not* vary (by design)

To isolate scale, we deliberately held constant: captions (schema and dropout), text tower (frozen Bio-ClinicalBERT), augmentations, image resolution, optimizer/schedule, and training budget/selection (EMA/SWA over epochs 16–20). Observed trends thus cannot be attributed to extra data, architecture changes, or per-size hyper-parameter tuning.

### 4.4 Limitations

Zero-shot OOD AUROCs are still close to chance on several cohorts. We did not evaluate calibration (ECE) or OOD detection, report confidence intervals, or perform per-cohort threshold transfer. Label harmonization across datasets used a unified but heuristic mapping (pathology keywords or BI-RADS $\geq 4$) that may not match local labeling protocols. We studied 2D images with minimal text; DBT and richer reports could further change the scaling curve. Finally, we used a single linear probe; few-shot target-domain adaptation may reveal steeper OOD scaling.

### 4.5 Implications and future work

Our results indicate that, within a fixed architecture and data regime, *parameter scale should be treated as a first-class design lever* for globally deployable mammography VLMs. Conversely, downsizing (smaller models, pruning, quantization) should be accompanied by OOD monitoring across diverse cohorts to avoid eroding low-FPR robustness. Future work will (i) add calibration and reject-option OOD detection, (ii) test few-shot adaptation curves vs. scale, (iii) expand to DBT and richer language supervision, and (iv) probe fairness slices (race, density, vendor, post-operative/implant cases) to understand whether scale interacts with equity objectives.

**Conclusion.** Holding architecture, data, and objective fixed, scaling a CLIP-trained VLM from ∼6M to ∼307M parameters produces consistent ID gains and *emergent* zero-shot OOD improvements concentrated at clinically relevant operating points. This scale-driven robustness—achieved without extra data or task labels—suggests that parameter count alone can act as a practical, deployment-relevant knob for improving global OOD behavior in multi-modal mammography models.

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

## A  APPENDIX

You may include other additional sections here.

