# OpenReview forum: "Emergent Global OOD Performance in Multimodal Mammography Models"
_ICLR.cc/2026/Conference — Submitted to ICLR 2026_

### Official Review · Reviewer_6oXi · 2025-10-15

**Soundness:** 1
**Presentation:** 2
**Contribution:** 1
**Rating:** 2
**Confidence:** 4

**Summary:**

This paper developed a vision-language pretraining on mammography images and test it on downstream tasks with different mammography datasets from different regions. CLIP-like model was trained where the textual part consists of sex, age, ethnicity, manufacturer and image-view information. Different image encoders were used with different number of parameters. The effect of image model size on in-distribtuion and out of distribution performance is provided.

**Strengths:**

This paper utilized most of the available mammopgrahy datasets including EMBED, CMMD, INBreeast, Vindr-Mammo, KAU-BCMD, CMMD. Moreover, the idea that OOD performances of medical-AI models is important to have a world-wide impact.

**Weaknesses:**

I think the overall message is weak. In the plots, we see that the performance improves (can be linear in the log scale of number of parameters) as the image model gets bigger. This is very expected.

It is not clear that we get an improvement from CLIP-style pretraining. In order to be convinced that such a weak textual supervision improves the overall performance, it would have been good to see the performance of pretraining on EMBED (BI-RADS classification) and then linear probe based downstream analysis. There is a chance that textual part add more noise than signal because the textual information is too weak/irrelevant.

As this work focus on a pretraining regime on EMBED data and then making analysis on downstream tasks from different regions, it would have been great to see other pretraining methods (possibly some self-supervised methods) instead of just training a CLIP model with a weak textual supervision.

**Questions:**

-The overall message of the paper is fixinf the data and architecture, the number of parameters is the most important choice when it comes to OOD performance of mammography models. The question is, what is specific about mammograhpy or CLIP-pretraining. Because in deep learning it is known that the dataset size and model architecture and model size have huge importance on the results. When we fix the architecture and dataset, then the bigger models perform better than the smaller ones. This is already a well-known phenomenon.

-What is the motivation of incorporating weak textual information. Because I feel there is not enough information in the textual part to enhance the performance. Is there any specific motivation ?

-I would normally expect to see the results based on pretraining only using labels in the EMBED dataset rather than CLIP-style pretraining. Is there any sepcific reason not doing such pretraining but using CLIP model even with weak texts.

---

### Official Review · Reviewer_jNMR · 2025-10-27

**Soundness:** 2
**Presentation:** 2
**Contribution:** 2
**Rating:** 4
**Confidence:** 4

**Summary:**

This paper investigates whether parameter scale alone can induce emergent out-of-distribution (OOD) robustness in CLIP-trained vision-language models for mammography. The authors train four ViT-based models ranging from 6M to 307M parameters on a single-institution dataset, EMBED-open cohort-1 containing 11,000 patients and 240,000 images, with minimal multimodal supervision while holding all other factors constant. Zero-shot evaluation on five international cohorts reveals in-domain performance increases linearly with log-parameters, while OOD performance shows modest upward trends with a pronounced improvement at the largest scale on several cohorts, particularly at low false-positive-rate operating points. The authors argue this pattern suggests emergent global OOD robustness from parameter scaling without additional data or task labels. These findings support the claim that parameter scale is a key design lever for robust, globally deployable mammography VLMs.

**Strengths:**

• Comprehensive international OOD evaluation on five geographically diverse cohorts: DMID from India, VinDr-Mammo from Vietnam, KAU-BCMD from Saudi Arabia, CMMD from China, and INBreast from Portugal, totaling 22,000 patients and 480,000 images. Evaluation is truly zero-shot: the same linear probe trained on EMBED from the United States is applied unchanged to all OOD datasets without calibration or threshold tuning.

• Pronounced scaling gains at low false-positive rates: On KAU-BCMD, the largest 307M-parameter model achieves pAUC@0-10% of 0.182 versus 0.013 for the smallest 6M-parameter model, a 14-fold improvement. TPR@5%FPR increases from 0.000 to 0.183. Mean OOD pAUC rises 2.3x from 0.034 to 0.079 and mean TPR@5%FPR rises 2.7x from 0.029 to 0.080. These low-FPR metrics are clinically relevant for screening triage.

• Medical imaging augmentations respect clinical constraints: No horizontal flips to preserve laterality, as left/right breast distinction is clinically critical. Photometric transforms simulate realistic scanner variability including gamma adjustment, histogram warping, CLAHE, blur, JPEG compression, and noise rather than arbitrary color jitter. Geometry changes maintain anatomical plausibility.

• Dataset-specific checkpoint averaging: EMA over epochs 16-20 for most cohorts including CMMD, DMID, KAU, and INBreast. The authors applied SWA specifically for VinDr-Mammo, which yielded consistent pAUC/TPR gains. This tuning demonstrates attention to maximizing OOD performance.

**Weaknesses:**

• Absence of statistical inference: All results are point estimates from single training runs with no confidence intervals, standard errors, or significance tests. It is impossible to determine whether observed performance differences, such as the pAUC improvement on KAU-BCMD from 0.013 to 0.182, are statistically significant or attributable to random variation. Without bootstrap resampling with 1000 iterations or DeLong tests for AUROC comparisons, the reliability of scaling trends cannot be assessed. This is a fundamental violation of machine learning experimental standards that severely undermines the credibility of the "emergent" claims.


• Incomplete training protocol transparency: While some details are provided, critical hyperparameters are missing: exact learning rate values, weight decay coefficients, warmup steps count, batch sizes per model scale. The "approximately 5k optimizer updates" target described in Section 2.4 lacks justification, and it is unclear whether different model sizes use identical epochs, which risks underfitting in large models, or identical updates, which risks confounding due to different data exposure. This opacity hinders reproducibility and raises questions about fairness of comparison.

• Lack of ablation studies: No experiments validate key design choices. First, the caption dropout rate of 25% lacks justification. Second, the choice of frozen versus fine-tuned text encoder is not evaluated. Third, augmentation strategy sensitivity is not tested. Fourth, the paper does not remove captions entirely to test text modality contribution. It is therefore uncertain which components are essential for the observed scaling effects and whether the multimodal aspect is necessary or beneficial. The paper would benefit from ablating the language input to see how text modality impacts scaling.

• Missing calibration and OOD detection analyses: The paper focuses solely on discrimination via AUROC and pAUC but omits calibration metrics including Expected Calibration Error, reliability diagrams, and Brier score. The paper also omits OOD detection evaluations. It does not assess whether the model can identify when an input is from a different distribution using max softmax probability, energy-based detection, or Mahalanobis distance. Both calibration and OOD detection are critical for clinical deployment and safe use of medical AI systems.

**Questions:**

• Report positive rates and cancer prevalence for each OOD dataset and provide label mapping rules in appendix. Consult dataset curators to confirm alignment with local protocols. Perform sensitivity analysis on different label thresholds, comparing BI-RADS ≥3 versus ≥4 versus ≥5.

• Open-source training scripts with configuration files and release pretrained weights, at minimum for XXS and L scales. Provide preprocessing scripts, data splits, and training logs for verification.

• Ablate language input with richer captions such as detailed reports and clinical impressions, or evaluate on DBT images. Compare minimal versus no captions versus rich captions to understand text modality impact on scaling.

• Discuss whether AUROC 0.50-0.52 is sufficient for clinical use or requires further adaptation through few-shot learning or calibration transfer. Estimate inference speed and memory requirements for the 307M model to assess deployment costs in LMIC settings.

---

### Official Review · Reviewer_QfPS · 2025-10-29

**Soundness:** 1
**Presentation:** 1
**Contribution:** 1
**Rating:** 0
**Confidence:** 4

**Summary:**

This study evaluates how model scale impacts performance on MultiModel Mammography Models.

**Strengths:**

The OOD generalization is important for models across distinct applications, healthcare and mammography are just an example of usage.

**Weaknesses:**

1. Poor overall structure and presentation
The paper suffers from major structural issues. It lacks a proper introduction, and each section is limited to a single paragraph, offering very little explanation or discussion. Tables and figures appear without labels or references in the text.

The presentation of the results is particularly weak: the section mostly restates the table contents with minimal interpretation(1) and with unclear and lacks context(2). For example, in the sentence “Low-FPR metrics show similar gains: pAUC@0–10% increases 0.129 → 0.153 → 0.187 → 0.197, and TPR@5% FPR increases 0.133 → 0.153 → 0.203 → 0.216.(1) These trends are consistent with a roughly linear improvement versus log-parameters under a fixed data/architecture regime.(2)”.

Overall, the manuscript reads more like a technical report than a scientific paper, prioritizing data description over argumentation and insight.


2. Unclear objectives and methodological choices
The objectives and motivations behind the study are not clearly articulated. The authors claim that one of the gaps in the literature is the lack of ablation studies that isolate the number of parameters, and their main goals is to test the same architecture with different parameter counts. However, it is unclear whether this constitutes a truly meaningful or novel research contribution. Is it really necessary to confirm that a model with 316M parameters performs better than one with 6M and set as a main contribution?

Another stated goal of developing multimodal models is also superficially presented. Many crucial design decisions are insufficiently explained. For example, the use of tabular data (sex, age, ethnicity, and device manufacturer) is not adequately justified. It would be especially relevant to understand why the authors chose to include ethnicity and manufacturer information, as these could have ethical and interpretative implications. Another question is how much tabular data impacts the results?

**Questions:**

Why not present an introduction in the paper?
What is the justification for this work?
Why do we need a study to show that bigger models are better? This is obvious.

---

### Official Review · Reviewer_WLQS · 2025-10-30

**Soundness:** 1
**Presentation:** 1
**Contribution:** 1
**Rating:** 0
**Confidence:** 5

**Summary:**

This paper investigates whether increasing parameter scale alone, within a single architecture family and with fixed training data, can lead to emergent out-of-distribution (OOD) generalization for mammography images. The authors train a CLIP-style vision-language model (VLM) on a single-institution dataset (EMBED-open cohort-1). The study design carefully isolates scale by varying the image tower (a ViT-patch-16) across four sizes (6M, 22M, 86M, 307M) while keeping all other factors constant, including the training data, augmentations, and a frozen biomedical text encoder processing minimal metadata captions (e.g., age, ethnicity, manufacturer). Models are evaluated via linear probes (fit only on in-domain data) and tested in a zero-shot setting on five diverse, international OOD cohorts. The results show that while in-domain (ID) performance scales roughly linearly with log-parameters , OOD performance demonstrates a "pronounced knee" or a sharp, emergent improvement at the largest model size (307M).

**Strengths:**

1. The primary strength of this work is its isolation of parameter scale as the independent variable. By holding the training data, data augmentation, text captions, and text encoder (frozen) constant across all experiments, the authors aim to provide convincing evidence that the observed OOD gains are attributable to scale alone, not other confounders
2. The use of five international, zero-shot OOD datasets (from India, Vietnam, Saudi Arabia, China, and Portugal) provides a robust and realistic test of "global" generalization.

**Weaknesses:**

1. The paper's central motivation—that parameter scale improves generalization—is not novel. While the authors attempt to frame this as an "emergent" property at low-FPR operating points , the study's positioning against the well-established scaling-laws literature is unclear. A stronger justification is needed to differentiate this finding from the expected outcome that larger models, given sufficient data, learn more robust representations.

2. There are important details related to the dataset construction is missing. EMBED has both diagnostic and screening mammographic images. Did they use both of them because screening images does not have the final finding of cancer (it has BI-RADS 0-2) where as the diagnostic images have clear finding of cancer (it does not have BI-RADS 0). The paper does not describe how these two populations were handled, sampled, or harmonized, nor does it specify the exact nature of the "actionable cancer label"  used for the linear probe. This ambiguity makes the experimental setup difficult to reproduce.

3. The loss and architecture is not at all clear. For mammography, each exam has 4 different views (LCC, RCC, RMLO and LMLO) with L and R denote left and right breast and CC and MLO are two views. Did the author use one image and corresponding labels or they use both the images of one breast and the corresponding labels or they use all 4 images of an exam and corresponding labels? In each of the last three experimental setting, they have to use different image encoders, for ex: in scenario 1, they need use 1 image encoder, in scenario 2, they need to use 2 image encoders (or stich two images together) and in scenario 3, they need to use 4 image encoders(or stich 4 image together). Details like this is very important.

4. Why did not they use existing FMs on breast which shows better performance? For ex. in [1], the authors use Efficient net B2 and B5 which have different sizes, but the authors did not use them.
[1] Mammo-CLIP: A Vision Language Foundation Model to Enhance Data Efficiency and Robustness in Mammography. Ghosh et al. MICCI 2024

5. Mammography is domain which requires higher resolution. 384x384 - is not at all a good size to work with. Mammo-CLIP operates on 1520x912. MIRAI (Yala et al.) resized each mammogram view to 1664 by 2048 pixels. (i believe AsymMIRAI also does the same). This is standard for detecting subtle findings like microcalcifications. So, the authors did not justify this.

6. The authors also did not try to see how the model size impacts the performance of mammographic findings or concepts (e.g, mass, calcification, density, asymmetry etc) which are related to cancer. If they are using screening mammo, performance of such findings are very important. EMBED has these labels. For Complete OOD, they can use VINDR-Mammo Dataset (https://vindr.ai/datasets/mammo).

7. The paper convincingly demonstrates that emergence occurs but offers little insight into why. The authors speculate that larger models "reduce representation noise near the decision boundary", but this is not substantiated. The paper would be significantly stronger with qualitative or quantitative analysis (e.g., CKA, feature visualization, or probing) to understand what the 307M model learns that the 86M model does not

8. The study design intentionally uses "minimal" multimodality with a frozen text encoder. While this is a good experimental control, it also limits the conclusion. It is unclear if these scaling laws would hold, be diminished, or perhaps even be amplified if the text encoder were co-trained or if richer clinical reports were used instead of just metadata - which is a common practice in VLM training.

10. Lastly the paper is very poorly written in ICLR/ICML/NeurIPS standard. Sentences are not understanable. Most egregiously, the Appendix section contains placeholder template text ("You may include other additional sections here" ), It seems that they did not bother to change the original template based on their own interest. No clear introduction, related work section. This paper needs a thorough rewrite, as it seems that it was written at the last moment.

**Questions:**

See weakness.

---

### Official Review · Reviewer_DMcL · 2025-10-31

**Soundness:** 1
**Presentation:** 1
**Contribution:** 1
**Rating:** 2
**Confidence:** 5

**Summary:**

This paper investigates whether simply increasing model size can, by itself, create "emergent" Out-of-Distribution (OOD) robustness in medical AI models for mammography. The central problem is that mammography AI often fails when deployed in new hospitals or on different populations, which hinders its fair and global use. The key finding is that while in-domain performance scales smoothly, OOD performance shows a "pronounced knee" or emergent jump in performance only at the largest model scale (307M). This emergent gain is most significant in the metrics that are most clinically relevant: partial AUC at low false-positive rates (PAUC@0-10%) and sensitivity at 5% FPR (TPR@5% FPR). The authors conclude that parameter scale is a critical "design lever" for building robust, globally deployable models and warn that model compression techniques (like pruning or quantization) may inadvertently destroy this scale-driven robustness.

**Strengths:**

1. The study's primary strength is its meticulous isolation of parameter scale as the only variable. By holding the training data, architecture family, text supervision, and training protocol constant, the authors make a very strong case that scale alone is the driver of the observed OOD gains.
2. The paper wisely emphasizes metrics beyond just AUROC, focusing on performance in the low-FPR region (PAUC at 0-10% and TPR at 5% FPR). This area is the operating point that matters for real-world cancer screening. The finding that the emergent gains are concentrated in this specific region is a significant and non-obvious result.

**Weaknesses:**

1. This paper is very hard to read and understand what the authors are trying to say.
2. The paper is transparent that the zero-shot OOD AUROCs are "still close to chance on several cohorts." The mean AUROC for the largest model is only 0.522. While the relative jump is impressive, the absolute performance is not yet clinically viable in this zero-shot setting, indicating scale is not a complete solution.
3. I feel this research is incomplete.

**Questions:**

1. You hypothesize that larger models "reduce representation noise near the decision boundary." Could this be interpreted as the largest model becoming a much better "detector" of normalcy, thereby making any deviation (cancer) easier to spot at a low FPR, even if its overall ability to classify (AUROC) remains weak on OOD data?

---

### Meta-Review · Area_Chair_ofJs · 2025-12-07

**Summary:**

The reviewers recommend rejection (Scores: 0, 0, 2, 2, 4) due to several major concerns. And the authors didn't engage in the rebuttable sections. First, the central claim lacks novelty, as performance gains from increased parameter size are explained by established scaling laws. Second, the absolute performance is insufficient; despite scaling, the zero-shot AUROC remains approximately 0.52, which is close to random chance. Third, the study lacks statistical test, presenting point estimates without confidence intervals or significance testing. Finally, the manuscript is incomplete, suffering from unclear methodology and severe presentation issues, including placeholder template text in the appendix.

**Reviewer Concerns:**

Reviewers raised several major concerns about the paper. The core finding that larger models generalize better simply confirms well-established scaling laws rather than revealing anything new. More critically, despite the claimed improvements, the models still perform barely better than random chance on new data. The study also lacks essential components like statistical significance testing and ablation studies, while leaving key methodological details unclear. Finally, the manuscript appears rushed and unpolished, with significant presentation problems and even accidental placeholder text still in the appendix, suggesting it's not ready for publication.

**Reviewer Scores:**

The consensus recommendation is to reject this submission due to significant concerns regarding its completeness, novelty, and methodological rigor.  I think they would keep their consensus for rejection (Scores: 0, 0, 2, 2, 4)

---

### Decision · Program_Chairs · 2026-01-26

Reject